# WHEN DOES DIFFUSION HELP? PDE-INSPIRED OPTIMIZATION ON FRAGMENTED AND NOISY DATA

**Rahul D Ray**
Department of Electronics and Electrical Engineering
Birla Institute of Technology and Science, Pilani – Hyderabad Campus
Hyderabad, India
`f20242213@hyderabad.bits-pilani.ac.in`

## ABSTRACT

Diffusion-inspired regularization, motivated by partial differential equations and smooth gradient flow, is increasingly used to stabilize neural network optimization. However, it remains unclear when such mechanisms materially affect learning, particularly under irregular data geometry and noisy supervision. We study diffusion-based optimization through a controlled experimental framework built on synthetic benchmarks that explicitly isolate geometric and statistical factors. We introduce a diffusion-regularized variant of stochastic gradient descent inspired by parabolic PDE smoothing and evaluate it on datasets exhibiting highly curved decision boundaries, disconnected supports, and varying levels of label noise. Across experiments, we analyze optimization dynamics, noise robustness, and loss landscape geometry under strictly matched training conditions. We find that diffusion regularization consistently smooths gradient flow and modifies local loss geometry, yielding stable convergence in fragmented regimes. However, improvements in predictive accuracy are strongly task-dependent and often dominated by dataset structure rather than optimizer choice. These results clarify when PDE-inspired diffusion meaningfully shapes optimization geometry, while highlighting its limitations as a general-purpose mechanism for improving performance.

## 1    INTRODUCTION

Optimization in deep learning is increasingly understood through the lens of continuous-time dynamics and partial differential equations. A line of work has shown that gradient-based training procedures can be interpreted as discretizations of underlying gradient flow or diffusion processes, providing a unifying explanation for techniques such as noise injection, annealed learning rates, and smoothing-based regularization Mobahi (2016); Elkabetz & Cohen (2021); Bah et al. (2022); Eberle et al. (2021). In this view, diffusion acts to suppress high-frequency components of the optimization trajectory, potentially altering the geometry of the loss landscape explored during training.

Parallel developments in diffusion models further strengthen the connection between diffusion processes and optimization. Guided diffusion has been shown to correspond to sampling solutions of regularized optimization problems, with theoretical guarantees under suitable assumptions Song & Ermon (2020); Guo et al. (2024). While these models are primarily generative, they reinforce the idea that diffusion is a principled mechanism for shaping optimization dynamics rather than merely a heuristic.

At the same time, a substantial body of work has investigated the geometry and connectivity of neural network loss landscapes. Empirical and theoretical studies have demonstrated that low-loss solutions found by gradient-based methods often lie in connected regions of parameter space, with smooth interpolation paths and shared basins of attraction Fort & Jastrzebski (2019); Gotmare et al. (2018); Kuditipudi et al. (2019); Shevchenko & Mondelli (2020); Horoi et al. (2022). These geometric properties help explain why modifications to optimization dynamics may change trajectory behavior and curvature without necessarily improving final accuracy. Related analyses of directional

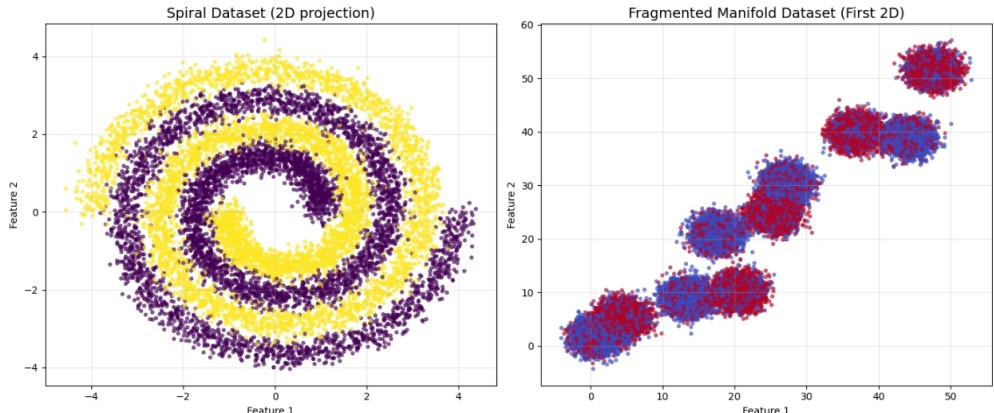

Figure 1: Two-dimensional projections of the synthetic datasets used in this study. **Left:** Spiral dataset, where samples from two classes form intertwined spiral arms, inducing a highly nonlinear but smooth decision boundary. **Right:** Fragmented manifold dataset, consisting of multiple disconnected clusters with locally coherent structure and globally non-smooth label assignments.

convergence and gradient alignment further indicate that optimization dynamics can converge in direction even when magnitudes and curvature differ Ji & Telgarsky (2020).

In scientific machine learning, particularly in physics-informed neural networks, PDEs are explicitly embedded into loss functions, transforming PDE solving into a challenging nonconvex optimization problem Cuomo et al. (2022); Nabian & Meidani (2019). Recent work has explored adaptive and PDE-aware optimization strategies in this setting, yet clear benefits over established optimizers such as Adam remain task-dependent Zhang (2018); Liu et al. (2019); Dereich et al. (2024); Ray (2025). Motivated by these insights, we study when diffusion-inspired optimization meaningfully influences learning by isolating geometric and statistical factors using controlled synthetic benchmarks.

## 2 CONTROLLED SYNTHETIC BENCHMARKS

Understanding when diffusion-inspired optimization influences learning requires experimental settings where geometric and statistical factors can be isolated and systematically varied. Real-world datasets often entangle heterogeneous noise, latent structure, and dataset-specific artifacts, obscuring the role of optimization dynamics. To address this, we construct controlled synthetic benchmarks that expose inductive biases and failure modes relevant to diffusion-based optimization. Our design employs two complementary datasets emphasizing distinct geometric regimes: a low-dimensional nonlinear setting with highly curved decision boundaries, and a high-dimensional regime with disconnected supports, globally non-smooth labeling rules, and explicit supervision noise. Across both datasets, geometry is controlled, labeling smoothness is varied, and noise is introduced in a principled manner to assess robustness independently of representational capacity.

### 2.1 SYNTHETIC DATASETS

We employ two complementary synthetic classification benchmarks designed to probe distinct geometric regimes relevant to diffusion-inspired optimization. The spiral dataset is a two-dimensional binary classification problem in which samples from two classes lie on intertwined spiral arms defined in polar coordinates, yielding smooth yet highly nonlinearly separable manifolds with strongly curved decision boundaries (Fig. 1, left). Isotropic Gaussian noise introduces localized perturbations while preserving global structure. The fragmented manifold dataset targets a substantially more challenging high-dimensional regime, with samples embedded in a 20-dimensional space and organized into ten well-separated Gaussian clusters (Fig. 1, right). Labels are assigned deterministically based on cluster parity, producing locally coherent but globally discontinuous supervision, with additional label noise injected to emulate annotation errors. Complete mathematical specifications and construction details for both datasets are provided in Appendix A

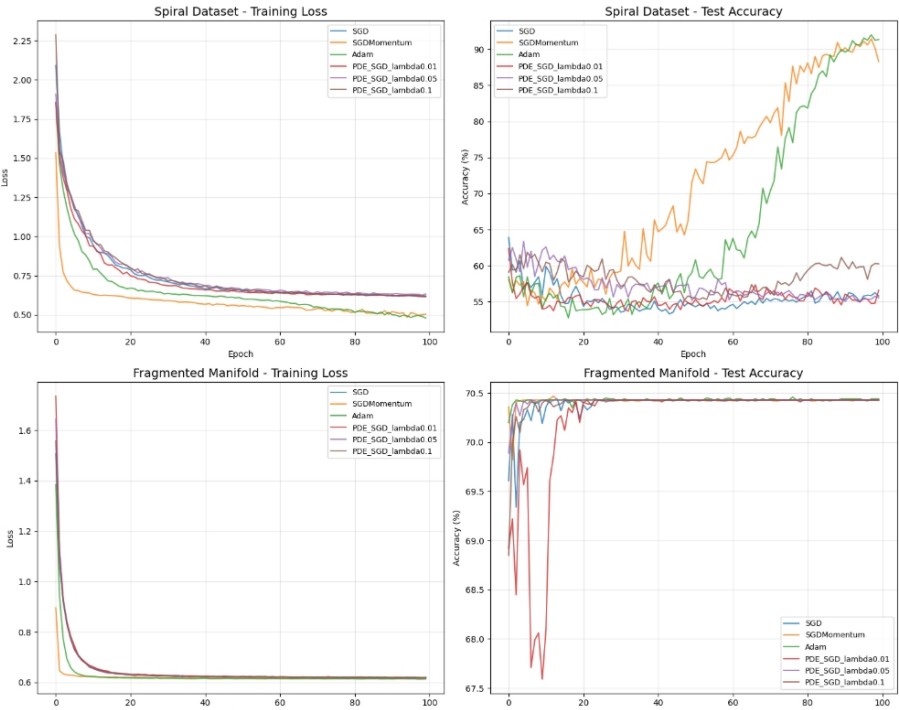

Figure 2: Training dynamics on synthetic benchmarks. **Upper left:** Training loss on the spiral dataset. **Upper right:** Test accuracy on the spiral dataset. **Lower left:** Training loss on the fragmented manifold dataset. **Lower right:** Test accuracy on the fragmented manifold dataset. All models are trained under identical initialization, batch size, and optimization settings, enabling a controlled comparison of convergence behavior across datasets with distinct geometric structure.

Together, these benchmarks isolate the geometric and statistical conditions under which diffusion-inspired optimization may alter optimization trajectories and loss landscape geometry, even when improvements in predictive accuracy are not guaranteed.

## 3 PDE-CONSTRAINED OPTIMIZATION AND MODEL ARCHITECTURES

### 3.1 PDE-CONSTRAINED OPTIMIZATION AND ARCHITECTURES

To investigate the effect of explicit diffusion regularization on neural network optimization, we introduce a PDE-constrained variant of stochastic gradient descent (SGD). This optimizer augments standard first-order updates with a diffusion term inspired by parabolic partial differential equations, with the objective of enforcing smoothness in gradient flow across neighboring parameters. Let $\theta_t \in \mathbb{R}^d$ denote model parameters at iteration $t$ and $L(\theta)$ the empirical loss. Standard SGD updates parameters according to

$$\theta_{t+1} = \theta_t - \eta \nabla L(\theta_t),$$

where $\eta > 0$ is the learning rate. In contrast, PDE-constrained SGD modifies the gradient via

$$\theta_{t+1} = \theta_t - \eta g_t, \qquad g_t = \nabla L(\theta_t) + \lambda \, \mathcal{L}(\nabla L(\theta_t)),$$

where $\lambda \geq 0$ controls the strength of diffusion and $\mathcal{L}$ denotes a discrete Laplacian over parameter space. This update can be interpreted as a forward Euler discretization of a parabolic PDE governing gradient flow, in which diffusion suppresses high-frequency gradient components while preserving coarse-scale descent directions, thereby encouraging smoother optimization trajectories.

Experiments are conducted using three neural network architectures spanning increasing levels of complexity, selected to evaluate PDE-constrained optimization across diverse learning regimes. These include fully connected and convolutional models applied to both synthetic and real-world

datasets. Details regarding Laplacian approximations, stability controls, optimizer implementation, and full architectural specifications are provided in Appendix B. Training dynamics on the synthetic benchmarks are summarized in Fig. 2.

## 4 EMPIRICAL EVALUATION

This section evaluates the proposed PDE-constrained optimization method on controlled synthetic benchmarks and a standard vision dataset. All experiments are conducted under strictly matched training conditions to isolate the effect of diffusion-based regularization from architectural and hyperparameter confounders.

### 4.1 SYNTHETIC BENCHMARKS AND LOSS LANDSCAPE GEOMETRY

We evaluate PDE-constrained optimization on two controlled synthetic datasets: a low-dimensional spiral dataset and a high-dimensional fragmented manifold dataset. For each dataset, a `SimpleMLP` architecture is trained using vanilla SGD, SGD with momentum, Adam, and PDE-constrained SGD with diffusion coefficients $\lambda \in \{0.01, 0.05, 0.1\}$. All models are trained for 100 epochs under identical initialization, batch size, loss function, and learning rate schedules. Figure 2 summarizes training loss and test accuracy dynamics. On the spiral dataset, vanilla SGD and PDE-constrained variants plateau near random-guess accuracy (approximately 55–60%), indicating limited ability to resolve highly curved decision boundaries. In contrast, SGD with momentum and Adam achieve substantially higher performance, reaching final accuracies of 88.3% and 91.35%, respectively, with Adam converging more slowly but attaining the highest final accuracy. On the fragmented manifold dataset, all optimizers converge rapidly to similar performance levels of approximately 70.4% test accuracy, suggesting that heavy label noise and global non-smoothness dominate generalization behavior. Consistent with these results, loss landscape analysis reveals smooth, unimodal interpolation paths between SGD and PDE-constrained solutions and similar local geometry across optimizers, with only modest differences in minima sharpness (Fig. 3). Additional analyses of noise robustness further confirm that performance degradation is governed primarily by dataset noise rather than optimizer choice (Appendix D).

## 5 DISCUSSION & CONCLUSION

This study examined diffusion-inspired optimization through a controlled empirical lens, prioritizing its effect on optimization dynamics rather than headline accuracy. Across synthetic benchmarks designed to isolate geometric and statistical factors, PDE-constrained SGD consistently altered gradient flow and local loss landscape geometry, producing smoother optimization trajectories and connected minima without inducing instability. However, these geometric modifications did not yield systematic improvements in predictive accuracy, particularly in regimes dominated by label noise or fragmented data supports. On low-dimensional manifolds with highly nonlinear decision boundaries, diffusion influenced convergence behavior but remained secondary to momentum-based and adaptive optimization methods. In high-dimensional noisy settings, dataset structure overwhelmingly governed generalization performance. Experiments on MNIST D.2 confirmed the scalability of PDE-constrained optimization while demonstrating no advantage over established optimizers. Overall, diffusion-based optimization emerges as a principled mechanism for shaping optimization geometry and stability, rather than a general-purpose approach for improving generalization.

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

# 6  APPENDIX

# A  ADDITIONAL DETAILS ON SYNTHETIC DATASET CONSTRUCTION

**Spiral dataset formulation.**  The spiral dataset is designed as a low-dimensional classification benchmark exhibiting strong nonlinear structure and highly curved decision boundaries. Let $N =$

10,000 denote the total number of samples, with equal class balance. For each class $c \in \{0, 1\}$, samples are indexed by $i \in \{1, \dots, N/2\}$ and generated in polar coordinates as

$$r_i^{(c)} = ai, \qquad \theta_i^{(c)} = bi + \phi_c,$$

where $a > 0$ controls the radial growth rate, $b > 0$ determines the angular frequency of the spiral, and the class-dependent offsets satisfy $\phi_1 - \phi_0 = \pi$, ensuring symmetric placement of the two spiral arms. The angular coordinate spans four full revolutions, producing expanding intertwined manifolds rather than concentric rings.

The polar coordinates are mapped to Cartesian space via

$$\tilde{x}_i^{(c)} = \begin{pmatrix} r_i^{(c)} \cos \theta_i^{(c)} \\ r_i^{(c)} \sin \theta_i^{(c)} \end{pmatrix}.$$

To introduce local perturbations while preserving the global geometric structure, isotropic Gaussian noise is added independently to each sample:

$$x_i^{(c)} = \tilde{x}_i^{(c)} + \epsilon_i, \qquad \epsilon_i \sim \mathcal{N}(0, \sigma^2 I_2),$$

with $\sigma = 0.2$. After generation, samples are randomly permuted and partitioned into 8,000 training and 2,000 test samples, maintaining exact class balance across splits.

**Fragmented manifold formulation.** The fragmented manifold dataset is constructed to model a substantially more challenging learning regime characterized by high dimensionality, disconnected supports, and globally non-smooth supervision. The dataset consists of $N = 50{,}000$ samples embedded in $\mathbb{R}^{20}$ and organized into $K = 10$ disjoint clusters.

Initial cluster centers are drawn as

$$\tilde{\mu}_k \sim \mathcal{N}(0, I_{20}),$$

and then explicitly separated by applying a cluster-dependent offset:

$$\mu_k = \tilde{\mu}_k + \alpha k v,$$

where $v \in \mathbb{R}^{20}$ is a fixed unit vector and $\alpha > 0$ enforces increasing inter-cluster separation. This construction ensures that clusters are well separated in feature space, preventing smooth interpolation between them.

Each cluster contributes an equal number of samples drawn according to

$$x_i^{(k)} \sim \mathcal{N}(\mu_k, \sigma_c^2 I_{20}), \qquad \sigma_c = 1.5.$$

Binary class labels are assigned deterministically based on cluster parity:

$$y_i^{(k)} = \begin{cases} 0, & k \text{ even}, \\ 1, & k \text{ odd}. \end{cases}$$

This labeling rule is locally smooth within clusters but globally discontinuous across cluster boundaries.

To emulate annotation noise, labels are independently corrupted at rate $\rho = 0.3$ via

$$\tilde{y}_i = \begin{cases} y_i, & \text{with probability } 1 - \rho, \\ 1 - y_i, & \text{with probability } \rho. \end{cases}$$

The dataset is shuffled and split into 40,000 training and 10,000 test samples with approximately balanced class proportions.

## B APPENDIX B: OPTIMIZER IMPLEMENTATION AND ARCHITECTURES

This section details the implementation of PDE-constrained stochastic gradient descent and the neural network architectures used across all experiments.

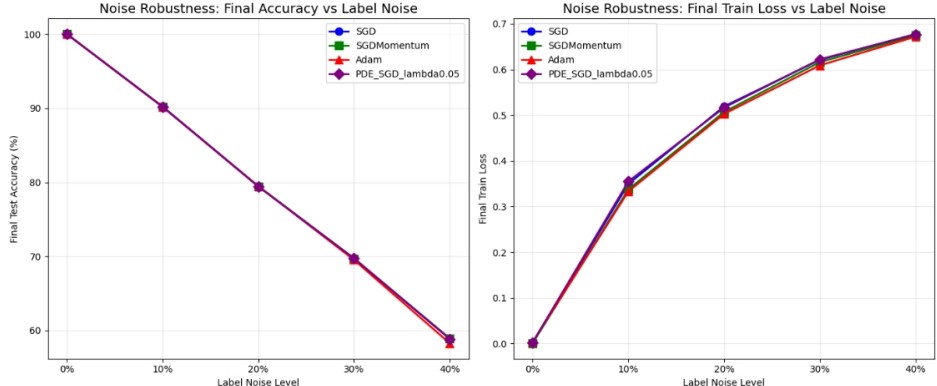

Figure 3: Noise robustness analysis on the fragmented manifold dataset. **Left:** Final test accuracy as a function of label noise level. **Right:** Final training loss as a function of label noise level. All models are trained under identical settings, isolating the effect of supervision noise on optimization and generalization behavior.

### B.1 DIFFUSION MECHANISMS

**Layer-based diffusion.** Gradients are smoothed independently within each parameter tensor. For a gradient tensor $g \in \mathbb{R}^n$, diffusion is approximated by removing the mean:

$$g_{\text{diff}} = g - \mathbb{E}[g], \qquad \mathbb{E}[g] = \frac{1}{n} \sum_{j=1}^{n} g_j.$$

This operation suppresses high-frequency intra-layer variation while preserving the dominant descent direction.

**Gradient-based diffusion.** All gradients are flattened into a single vector $g \in \mathbb{R}^d$. Diffusion is applied using a one-dimensional finite-difference Laplacian:

$$(\mathcal{L}g)_i = g_{i+1} - 2g_i + g_{i-1}.$$

The diffused gradient is reshaped back into the original parameter tensor structure prior to the update.

### B.2 LAPLACIAN APPROXIMATION AND STABILITY CONTROLS

Constructing an exact graph Laplacian over modern neural networks is computationally infeasible due to the scale and heterogeneity of parameter tensors. We therefore employ efficient Laplacian approximations that preserve the qualitative behavior of diffusion while remaining tractable.

Across all variants, optional $\ell_2$ weight decay is applied directly to the gradient prior to diffusion, and global gradient norm clipping is used to ensure numerical stability. Let $g_t$ denote the combined gradient. Clipping is performed as

$$\tilde{g}_t = \begin{cases} g_t, & \|g_t\|_2 \leq \tau, \\ \tau \dfrac{g_t}{\|g_t\|_2}, & \|g_t\|_2 > \tau, \end{cases}$$

where $\tau > 0$ is a fixed threshold.

When momentum is enabled, updates follow the classical heavy-ball formulation,

$$v_{t+1} = \mu v_t + \tilde{g}_t, \qquad \theta_{t+1} = \theta_t - \eta v_{t+1},$$

with momentum coefficient $\mu \in [0, 1)$. Parameter updates are applied in a no-gradient context to ensure computational correctness.

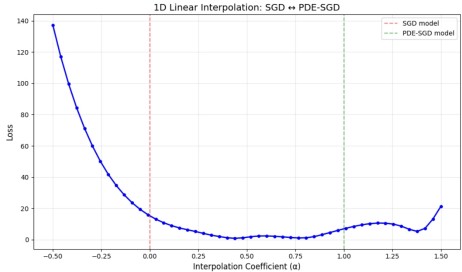

Figure 4: Loss evaluated along a linear interpolation path between solutions obtained using standard SGD and PDE-constrained SGD on the fragmented manifold dataset. The interpolation coefficient $\alpha$ parametrizes the convex combination of model parameters, with $\alpha = 0$ corresponding to the SGD solution and $\alpha = 1$ to the PDE-constrained solution.

## C    IMPLEMENTATION DETAILS

The proposed optimizer is implemented as a subclass of the PyTorch Optimizer interface, enabling seamless integration with standard training pipelines and existing model architectures. All hyperparameters, including the learning rate, diffusion coefficient, momentum factor, and weight decay, are explicitly validated to ensure non negativity and numerical stability prior to training. Default values are chosen to match those of standard SGD where applicable, allowing controlled comparisons across optimization methods.

Gradient diffusion is applied directly to the parameter gradients at each optimization step, following the formulation described in Section 3. To ensure computational stability, optional global gradient norm clipping is supported, and parameter updates are executed in a no gradient context. When momentum is enabled, updates follow the classical heavy ball formulation, with diffusion applied to the instantaneous gradient before velocity accumulation.

Functional correctness of the implementation was verified through unit tests on simple multilayer perceptrons, including forward and backward passes, convergence checks under known loss functions, and consistency tests against vanilla SGD when the diffusion coefficient is set to zero. Additional sanity checks confirm that the optimizer reproduces standard SGD behavior in the appropriate limit and introduces no unintended side effects in gradient computation or parameter updates.

### C.1    NEURAL NETWORK ARCHITECTURES

**SimpleMLP.**    A three-layer fully connected network with two hidden layers of 128 units, batch normalization, ReLU activations, dropout, and a linear output layer. Kaiming normal initialization is used throughout.

**MediumMLP.**    A five-layer fully connected network for flattened image inputs such as MNIST, with progressively decreasing feature dimensionality and interleaved batch normalization and dropout.

**SimpleCNN.**    A compact convolutional network with two convolutional blocks, batch normalization, max pooling, and a fully connected classifier head. Feature dimensionality is computed dynamically to support both grayscale and RGB inputs, with dropout applied in the penultimate layer. All architectures were verified to execute correctly on GPU hardware.

## D    APPENDIX D: EXTENDED ANALYSES

### D.1    NOISE ROBUSTNESS

We next assess robustness to increasing label noise on the fragmented manifold dataset, varying the noise rate from 0% to 40%. Models are trained using SGD, SGD with momentum, Adam, and PDE-constrained SGD with $\lambda = 0.05$ under identical

As shown in Fig. 3, performance degrades monotonically with increasing noise across all optimizers. While PDE-constrained SGD produces smoother gradient magnitude trajectories, it does not yield consistent accuracy improvements. Robustness, quantified via the area under the accuracy–noise curve, is nearly identical across methods, indicating that dataset noise rather than optimizer choice governs performance in this regime.

## D.2 MNIST SCALE-UP

To evaluate scalability, we conduct experiments on MNIST using a fully connected `MediumMLP` and a convolutional `SimpleCNN`. Models are trained for 30 epochs using SGD, SGD with momentum, Adam, and PDE-constrained SGD with $\lambda \in \{0.01, 0.05\}$.

For `MediumMLP`, Adam achieves the highest accuracy (98.61%), followed by SGD with momentum (98.36%). PDE-constrained variants match vanilla SGD at approximately 96.3% accuracy with slightly increased training time. For `SimpleCNN`, all optimizers exceed 98.7% accuracy, with Adam and momentum-based SGD performing best. PDE-constrained SGD again matches standard SGD but does not surpass adaptive methods.

## D.3 LOSS LANDSCAPE GEOMETRY

Two-dimensional loss contours and three-dimensional loss surface visualizations around Adam-trained solutions reveal smooth basins with gradual curvature. Minima sharpness is evaluated using random perturbations of magnitude $\pm 0.1$. Adam and momentum-based SGD exhibit the lowest sharpness values, while PDE-constrained SGD shows slightly higher values on the same order of magnitude ($\sim 10^{-4}$), indicating modest geometric differences without qualitative changes.

