# OpenReview forum: "When Does Diffusion Help? PDE-Inspired Optimization on Fragmented and Noisy Data"
_ICLR.cc/2026/Workshop/Sci4DL — Sci4DL 2026_

### Official Review · Reviewer_Gn5D · 2026-02-14

**Fit:** 2
**Significance:** 2
**Confidence:** 2

**Summary:**

The submission studies whether adding diffusion-style smoothing to the optimization dynamics can help the learning outcomes. To this end, the authors propose a PDE-constrained SGD variant that smooths gradients via a discrete Laplacian term.

**Strengths:**

The paper is well-motivated and provides a clear empirical framing. It is intuitive that diffusion regularization is best viewed as a principled stabilizer / geometry-shaper for optimization.

**Suggestions:**

There are some typos in the paper.
e.g.:
1. "does not surpass adaptive methods. conditions."


Further, It would be more interesting to visualize the loss landscape.
Li, Hao, et al. "Visualizing the loss landscape of neural nets." Advances in neural information processing systems 31 (2018).

---

### Official Review · Reviewer_Ypuc · 2026-02-27

**Fit:** 3
**Significance:** 2
**Confidence:** 3

**Summary:**

This paper studies an optimization scheme inspired by parabolic PDE smoothing. The proposed scheme adds a discrete Laplacian over parameter space to the standard SGD updates which mimics the numerical scheme of a parabolic PDE. This comes with an extra hyper parameter $\lambda$ which represents the strength of the diffusion term. The authors claim that this should modify the effective loss landscape by suppressing high-frequency gradient components. It studies its performances in two controlled settings with synthetic data: one with curved decision boundaries and another with disconnected supports. The author compares it with other classical optimization method such as SGD, SGD+momentum and Adam. The authors also used different architectures such as fully connected and convolutional models. PDE inspired scheme has the same performances as SGD with no momentum. They demonstrate a negative result: their proposed optimizer does not surpass existing schemes.

**Strengths:**

- Provides a new optimization method inspired by parabolic PDEs.
- The central question of the paper is well-motivated.
- Conducts a thorough analysis and comparaison with existing schemes (vanilla SGD, SGD with momentum, Adam) on different datasets (synthetic benchmarks and MNIST) and architectures (MLP, CNN).

**Suggestions:**

- Have the authors considered evaluating on simpler datasets or other task
- Have the authors explored parabolic PDE + momentum?
- Appendix D.3.: can the authors explain the difference between the geometry of the minima reached by Adam / momentum based SGD and the ones reached by PDE-constrained SGD? The properties of the minima reached by this new scheme would be an interesting extension of this work.
- The authors keep the hyper parameters the same for all the different schemes but maybe this new scheme needs hyperparameters tuning to perform well. Have the authors tried to change the learning rate for instance?
- Regarding presentation, the scheme is introduced only in Section 3. Moving it earlier would improve the clarity of the paper.
Typos:
- Typo l.436 ".conditions"
- Figure 4 caption and Appendix D.3 text appear misaligned.

---

### Meta-Review · Area_Chair_xCuM · 2026-03-01

**Recommendation:** Accept

**Metareview:**

Based on the reviews, I recommend accept.

---

### Decision · Program_Chairs · 2026-03-02

Accept